# The Adhesion G-Protein-Coupled Receptor GPR115/*ADGRF4* Regulates Epidermal Differentiation and Associates with Cytoskeletal KRT1

**DOI:** 10.3390/cells11193151

**Published:** 2022-10-07

**Authors:** Romy Winkler, Marianne Quaas, Stefan Glasmacher, Uwe Wolfrum, Torsten Thalheim, Jörg Galle, Knut Krohn, Thomas M. Magin, Gabriela Aust

**Affiliations:** 1Research Laboratories and Clinic of Orthopedic Surgery, Traumatology and Plastic Surgery, Leipzig University and University Hospital, 04103 Leipzig, Germany; 2Research Laboratories and Clinic of Visceral, Transplantation, Thoracic, and Vascular Surgery, Leipzig University and University Hospital, 04103 Leipzig, Germany; 3Institute of Molecular Physiology, Molecular Cell Biology, Johannes Gutenberg University of Mainz, 55128 Mainz, Germany; 4Interdisciplinary Center for Bioinformatics (IZBI), Leipzig University, 04107 Leipzig, Germany; 5Core Unit DNA-Technologies, Leipzig University, 04103 Leipzig, Germany; 6Division of Cell and Developmental Biology, Institute of Biology, Leipzig University, 04103 Leipzig, Germany

**Keywords:** GPR115, epidermis, keratinocyte, KRT1

## Abstract

Among the 33 human adhesion G-protein-coupled receptors (aGPCRs), a unique subfamily of GPCRs, only *ADGRF4*, encoding GPR115, shows an obvious skin-dominated transcriptomic profile, but its expression and function in skin is largely unknown. Here, we report that GPR115 is present in a small subset of basal and in most suprabasal, noncornified keratinocytes of the stratified epidermis, supporting epidermal transcriptomic data. In psoriatic skin, characterized by hyperproliferation and delayed differentiation, the expression of GPR115 and KRT1/10, the fundamental suprabasal keratin dimer, is delayed. The deletion of *ADGRF4* in HaCaT keratinocytes grown in an organotypic mode abrogates *KRT1* and reduces keratinocyte stratification, indicating a role of GPR115 in epidermal differentiation. Unexpectedly, endogenous GPR115, which is not glycosylated and is likely not proteolytically processed, localizes intracellularly along KRT1/10-positive keratin filaments in a regular pattern. Our data demonstrate a hitherto unknown function of GPR115 in the regulation of epidermal differentiation and KRT1.

## 1. Introduction

The epidermis, a stratified epithelium, continuously renews from basal keratinocyte stem cells that migrate outwards to give rise to differentiated keratinocytes that ultimately form the skin barrier. In healthy epidermis, proliferation is restricted to keratinocytes in the basal layer, where the various stem cell types reside [1]. Their progeny constantly undergo differentiation to form the upper suprabasal layers, the stratum spinosum and stratum granulosum, and ultimately the stratum corneum. In normal skin, keratinocyte proliferation and differentiation are highly regulated to maintain epidermal homeostasis. The disruption of the underlying regulatory circuits contributes to conditions such as psoriasis, which is characterized by hyperproliferation and delayed differentiation [2]. Epidermal cytoarchitecture and tissue cohesion largely depend on the expression of keratins, which form the cytoskeletons of all epithelia, including the epidermis. Keratins protect the epidermis against mechanical stress, secure intercellular adhesion, and act as signaling hubs via multiple protein interactions [3,4]. They form extensive filament arrays through the heterodimerization of type I and type II keratins, which assemble into bundles of intracellular filaments. Keratin expression is tightly regulated to support keratinocyte-specific functions. Basal skin keratinocytes express KRT5 and KRT14 (KRT5/14), whereas differentiated suprabasal keratinocytes switch off KRT5/14 to be replaced by KRT1/10, which serve as major constituents of the cornified envelope [5]. How epidermal differentiation and keratin expression are regulated remains incompletely understood.

Adhesion G-protein-coupled receptors (aGPCRs), a unique subfamily of GPCRs [6,7], participate in a myriad of diverse processes such as immune regulation, brain development, cell positioning, metabolism, and tumorigenesis [8]. They consist of a large N-terminal extracellular domain (ECD) connected to the seven-span transmembrane (7TM) helices, followed by the intracellular domain (ICD). The ECD contains consecutive adhesive folds, facilitating adhesion, and the juxtamembranous GPCR autoproteolysis-inducing (GAIN) domain, which harbors the GPCR autoproteolysis site (GPS). The cleavage of most aGPCRs into two noncovalently associated fragments at this site is usually a prerequisite for their activity. Impaired self-cleavage is attributed to alterations in the consensus GPS sequence [9,10].

Among all human aGPCRs, GPR115, encoded by *ADGRF4* (in the following sections we use GPR115 for the protein and *ADGRF4* for the gene), is unique in displaying a marked skin-dominated transcriptomic profile [11]. In fact, *ADGRF4* is among the 100 genes (39th place) that exhibit the highest mRNA expression in the skin, according to analyses of 100 different tissues and cell types [12]. Given the limited knowledge about GPR115 expression and function in the skin, we set out to investigate its expression, interaction, and function. Following epidermal single-cell RNA sequencing (scRNAseq) analyses suggesting that GPR115 protein should correlate with a rare subset of basal and most suprabasal keratinocytes, we verified its expression in these compartments. Further, we demonstrated that a loss of *ADGRF4* reduces epidermal stratification in human keratinocytes grown in organotypic mode and abrogates *KRT1*, indicating a role of GPR115 in epidermal differentiation.

## 2. Materials and Methods

### 2.1. Ethics Statement

The Ethics Committee of the Medical Faculty of Leipzig University approved the study of human skin for aGPCR expression (no. 386/18). All patients gave informed consent.

### 2.2. Antibodies (Abs) and Plasmids

The Abs used are summarized in Appendix A. *ADGRF4*, amplified from human keratinocytes by RT-PCR, was cloned into the pcDNA3.1 plasmid. mGPR115 pcDps, encoding mouse GPR115, was kindly provided by I. Liebscher (Leipzig University).

For detection purposes, an HA tag was inserted directly downstream of the signal peptide, and a Flag tag was inserted at the C-terminus of the receptor using the Q5^®^ Site-Directed Mutagenesis Kit (New England Biolabs GmbH, Frankfurt, Germany). The constructs encoding GPR115 with mutated glycosylation sites (GPR115 mutGlyc and GPR115 noGlyc) were purchased from OriGene Technologies GmbH (Herford, Germany). All created mutations were confirmed by sequencing. The plasmids were transfected into suitable cells using Lipofectamine 2000 (Thermo Fisher Scientific, Darmstadt, Germany).

### 2.3. Culture of Primary Cells and Cell Lines

HaCaT [13] (CLS GmbH, Eppelheim, Germany), Cos-7 (DMSZ GmbH, Braunschweig, Germany), and NIH-3T3 cells (LGC Standards GmbH, Wesel, Germany) were cultured in DMEM/10% fetal calf serum. For the N-glycosylation analysis, cells were incubated with 5 µg/mL tunicamycin (Merck KGaA, Darmstadt, Germany) for 30 h before lysis. Mouse wild-type (WT) keratinocytes and keratinocytes deficient of type I keratins (*KtyI^-/-^*) were cultured as described in [14].

### 2.4. GPR115 Knockout (KO) HaCaT Clones

Single guide RNAs (sgRNAs) for *ADGRF4* were designed (www.e-crisp.org/E-CRISP/designcrispr) (accessed on 1 February 2018). sgRNA1 (ggtgaatcttggatctatag) and sgRNA2 (ctatagatccaagattcacc) targeted exon 2 and 6, respectively. pSpCas9BB-2A-GFP PX458, a gift from Feng Zhang (Addgene plasmid #48138), was digested with BsmB1 and ligated with annealed sgRNAs [15]. HaCaT cells were transfected with these plasmids. After 24 h, GFP-positive cells were sorted (FACSAria SORP, Becton Dickinson, Heidelberg, Germany), cultured as a monolayer, recloned, and stained for GPR115. Clones of both *ADGRF4* sgRNAs with decreased numbers of GPR115-positive cells were selected and named according to the applied sgRNA as GPR115KO1 and -KO2. gDNA sequencing of all clones confirmed the mutation of the target loci. Mutations partly hit both alleles differently and are multifaceted. RNA sequencing was performed as recently described in [16].

### 2.5. Organotypic Skin Constructs

NIH-3T3 fibroblasts were treated with 4 µg/mL mitomycin C (Merck) for 2 h. Then, 4 × 10^4^ postmitotic fibroblasts were embedded in 400 µL of rat tail collagen type I (Corning GmbH, Kaiserslautern, Germany) in a cell culture insert used in a 12-well carrier plate. After 24 h, the gel was coated with 5 µg/mL fibronectin (Merck) for 1 h, overlayered with 4 × 10^5^ HaCaT cells, and cultured for 10 days. Constructs were fixed in phosphate-buffered 4% formaldehyde for 45 min and embedded in Tissue Tek (Thermo Fisher).

### 2.6. Crystal Violet Assay

To quantify the cell numbers of HaCaT clones, 1 × 10^3^, 2 × 10^3^, and 4 × 10^3^ cells/well (three wells/density) were seeded in a 96-well plate and cultured for 48 h. The adherent cells were fixed for 10 min in 96% ethanol then washed and stained with 0.5% crystal violet/20% methanol for 25 min. After washing, the cells were lysed overnight with 100 μL of 0.5% Triton X-100. The absorbance was measured at 590 nm.

### 2.7. Immunolabeling and Proximity Ligation Assay (PLA)

Attached cultured cells or 6 µm tissue cryosections were fixed with ice-cold acetone for 10 min and immunostained as described in [16]. The Abs used for labeling are listed in Appendix A. To detect and localize the specific interactions of endogenous proteins, the DuoLinkTM in situ PLA (Olink Bioscience, Uppsala, Sweden) was performed [17]. All immunofluorescence stains were imaged by confocal laser scanning microscopy (LSM800 or LSM880 fast Airyscan; Carl Zeiss AG, Oberkochen, Germany).

### 2.8. Immunoelectron Microscopy

Postembedding immune-gold labeling was performed as described in [18]. Briefly, human skin samples were fixed in 0.1 M phosphate buffer with 0.1% glutaraldehyde and 4% paraformaldehyde. Samples were embedded in LR White resin (Merck) and polymerized at 4 °C via UV radiation. Ultrathin sections were stained with the GPR115^ECD^ Ab in combination with the secondary Ab conjugated with NanogoldTM (Nanoprobes, Yaphank, NY, USA). For subsequent silver enhancement [19], the particle size was adjusted by the duration of incubation. Sections counterstained with ethanolic uranyl acetate underwent an analysis using an FEI Tecnai12 BioTwin TEM equipped with an SIS Mega-View3 CCD camera (Olympus Soft Imaging Solutions GmbH, Muenster, Germany). The images were analyzed using Fiji.sc.

### 2.9. Flow Cytometry

Cell surface expression was quantified by flow cytometry. Cells were incubated with the primary Ab for 30 min, followed by a fluorophore-labeled secondary Ab for 20 min. Cells were fixed with 1% paraformaldehyde/PBS and analyzed in an FACSAria III.

### 2.10. Western Blot Analysis

To isolate the epidermis, skin samples were incubated in PBS for 2 min at 60 °C. The epidermis was pulled off and stored in liquid nitrogen for further applications. Cells and epidermal samples were lysed either in 6.5 M urea buffer (50 mM Tris pH 7.4, 1 mM EGTA, 6.5 M urea, 2 mM DTT), RIPA lysis buffer, or M-PER (Thermo Fisher), each containing Halt™ protease and a phosphatase inhibitor cocktail (Thermo Fisher). For the glycosylation analysis, lysates were incubated with PNGase F (NEB GmbH, Frankfurt, Germany). The Western blot analysis was performed as described in [20].

### 2.11. Single-Cell RNA Sequencing (scRNAseq) Data Reanalysis

Data (GSE147482) [1] were downloaded from the GEO database. Using 300 marker genes for basal cell types I-IV (BAS I-IV), spinous types I-II (SNP I-II), and granular (GRN) keratinocytes as well as melanocytes (MEL) identified by Wang et al. [1] (library clustering), we ran a self-organizing map (SOM) analysis of all cells to identify cells with specific expression profiles [21]. The SOM spot analysis allowed the identification of cells that significantly overexpressed BAS I-III, GRN, and MEL signatures. The BAS IV and SPN I-II signatures were rather broadly distributed. A cluster of cells overexpressing these three signatures together and a part of the GRN signature was labeled as ‘MIX’. Violin plots of the expression of individual genes were generated using the R-package Seurat 4.1.1 [22].

### 2.12. Statistical Analysis

Means ± SEM are given. Differences between HaCaT WT and GPR115 clones were analyzed by the Mann–Whitney U test. *p* values < 0.05 were considered significant.

## 3. Results

### 3.1. High ADGRF4 Levels in Rare Basal and in Most Suprabasal Keratinocytes

*ADGRF4* shows a distinct skin-dominated transcriptomic profile when comparing different human tissues (Figure 1a) [11]. To define this more precisely, we reanalyzed scRNAseq data of human neonatal epidermis [1]. We identified three subsets of basal keratinocytes (Figure 1b). The *ADGRF4* level was low in most type-III basal cells, characterized by high *COL17A1*, encoding a structural component of hemidesmosomes. Notably, these are stem cells with an unlimited capacity for self-renewal that are generally quiescent. Most type-I and -II basal cells proliferate; they are characterized by high levels of cell-cycle-related *CDC20*. The type-II restricted expression of *UHRF14*, playing a role in the G1/S phase transition, discriminates this population from type-I basal cells, and 14.5% and 20.3% of type-I and -II basal cells, respectively, had high *ADGRF4* levels. Thus, GPR115 is likely expressed in a rare subset of basal keratinocytes. High *ADGRF4* was present in 71.0% of the cells in the MIX cluster, including spinous keratinocytes, and in almost all granular keratinocytes. Consistent with the predominant *ADGRF4* in differentiated keratinocytes, *DSC1*, encoding the desmosomal protein desmocollin-1, and *IV*L, encoding involucrin starting to be synthesized in the stratum spinosum, showed profiles similar to that of *ADGRF4*. The closest relative of *ADGRF4*, *ADGRF2*, was almost absent in human keratinocytes. Melanocytes, positive for the specific transcription factor *MITF*, were *ADGRF4*-negative.

### 3.2. GPR115 Shows Intracellular Distribution in Human Epidermis

To validate these transcriptomic data at the protein level, we first carefully characterized GPR115 antibodies (Abs). We applied HaCaT wild-type (WT) keratinocytes and a HaCaT GPR115 knockout (KO) clone in which most *ADGRF4* transcripts were lost as well as Cos-7 cells expressing HA-tagged GPR115 (Appendix A). We focused on the GPR115^ECD^ Ab, which binds the receptor at its ECD and stained 5–8% of the HaCaT WT cells but almost none of the *ADGRF4*-deficient GPR115KO2/2 cells. In HA-GPR115 Cos-7 cells, a one-to-one overlapping of the Tag with GPR115^ECD^ Ab labeling was seen.

Applying this Ab on cryosections of human epidermis, we detected GPR115 in few basal and in almost all KRT10-positive suprabasal, noncornified keratinocytes (Figure 1c), which is in full agreement with the transcriptomic profile. We noted that basal GPR115-positive cells touched the basal membrane only with little membrane extensions but not across their entire basal site (Figure 1d,e). Unexpectedly, GPR115 displayed an apparent intracellular distribution in keratinocytes in vivo and lacked the expected membrane-associated localization pattern (Figure 1d). To further control GPR115^ECD^ Ab specificity, epidermal staining was repeated after the preincubation of the Ab with Cos-7 cells expressing GPR115, which strongly reduced intracellular epidermal staining (Appendix A).

In addition to the interfollicular epidermis, GPR115 expression was examined in skin appendages such as sebaceous and sweat glands and hair follicles (Appendix A), where it showed a very similar distribution as KRT1 and KRT10 (Appendix A).

### 3.3. Delayed GPR115 Expression in Suprabasal Psoriatic Keratinocytes

Our in vivo data are compatible with an involvement of GPR115 in epidermal differentiation. To obtain further insight, we turned to psoriatic skin as a model for disturbed epidermal homeostasis. In lesional psoriatic skin, characterized by excessive keratinocyte proliferation and altered differentiation, the distribution of GPR115 and keratins was changed, especially in rete ridges (Figure 2a,b). The expression of GPR115 and KRT1/10 was delayed until keratinocytes reached the granular layer. Simultaneously, many suprabasal cells expressed the proliferation marker Ki-67 (Figure 2b). Of these Ki-67^+^ keratinocytes, ~25% were positive for GPR115. In granular keratinocytes outside the rete ridges, GPR115 and KRT1 are always present but consistently appear non-homogenously in various cells.

To address whether GPR115 preceded KRT1 or vice versa, immunostaining for both proteins was performed, focusing on the first GPR115-positive or KRT1-positive cells upon leaving the basal layer (Figure 2c). The notion that GPR115^+^ KRT1^−^ but no GPR115^−^ KRT1^+^ keratinocytes exist supports the hypothesis that GPR115 precedes the expression of KRT1. To investigate this, we turned to HaCaT-based organotypic skin equivalent models and performed loss of function studies.

### 3.4. Loss of ADGRF4 Negatively Regulates KRT1 Expression and Epidermal Differentiation

In 2D cultures, HaCaT keratinocytes formed monolayers in which 5–8% of the cells expressed KRT1/10, consistent with previous findings [13], and GPR115 (Figure 3a,b). Notably, when the experiment was repeated with GPR115KO HaCaT clones, the percentage of KRT1/10-positive cells declined obviously (Figure 3a,b), caused by a strong, up to 220-fold decrease in KRT1 transcripts compared with HaCaT WT cells (Figure 3c). Notably, the level of remaining *KRT1* transcripts and the percentage of KRT1/10-positive cells in the various clones were correlated. The moderate up to 12-fold reduction in *KRT10* in GPR115KO HaCaT clones is compatible with the individual, but not pairwise, regulation of type I and type II keratins [23].

Upon the 3D organotypic growth of HaCaT WT cells, schematically shown in Figure 3d, GPR115 was strongly induced in nearly all suprabasal cells, whereas most basal cells were GPR115-negative, as in a normal epidermis (Figure 3e,f). Remarkably, in HaCaT GPR115KO clones, the suprabasal cells remained negative for KRT1/10 (Figure 3e), indicating their likely inability to differentiate, and the keratinocyte layer formation looked disturbed.

To investigate this further, the number of keratinocyte layers in organotypic-grown HaCaT WT or GPR115KO clones was quantified (Figure 3g). The loss of *ADGRF4* significantly decreased the number of suprabasal, differentiated epidermal layers, and thus the number of epidermal cells in organotypic constructs built from these two clones showed the lowest *ADGRF4* levels (Figure 3h). To exclude that these differences between HaCaT WT cells and GPR115KO clones might be due to a direct regulation of proliferation by GPR115, their cell numbers were quantified in monolayer cultures where no difference was found (Figure 3i). Thus, HaCaT WT cells proliferate more compared to the GPR115KO clones under organotypic conditions but not under monolayer conditions. In sum, our data strongly indicate that GPR115 promotes epidermal differentiation and regulates KRT1.

### 3.5. Endogenous GPR115 Colocalizes with KRT1/10-Positive Filaments

In human epidermis, GPR115 was localized inside keratinocytes (Figure 1d). A highly similar distribution was found in immunostained monolayered HaCaT keratinocytes (Figure 4a). Flow cytometry confirmed the intracellular localization of endogenous GPR115; HaCaT keratinocytes did not show GPR115 at the cell surface (Figure 4b). In contrast, Cos-7 cells transfected with GPR115 pcDNA3.1 localized GPR115 along the cell surface. One major difference between Cos-7 and HaCaT cells relates to the set of keratins they express. While the former are characterized by the simple epithelial keratins KRT8/18, the latter express KRT5/14 under proliferative and KRT1/10 under differentiation conditions.

To exclude the possibility that only a fragment of the GPR115 ECD locates intracellularly or that a C-terminal fragment might escape detection by the GPR115^ECD^ Ab, an Ab binding to the ICD (GPR115^ICD^) was applied (Appendix A). This Ab confirmed the former results (Figure 4c). Consistently, both Abs yielded an intracellular, filament-like pattern resembling that of cytoskeletal keratins (Figure 4a,c). Costaining with keratin-specific Abs showed a strong overlap of GPR115 with KRT1/10. All GPR115-positive cells expressed KRT1, and most were also positive for KRT10 (Figure 4d). To independently probe for this unexpected association of an aGPCR and a cytoskeletal keratin, a PLA was performed for endogenous proteins in HaCaT keratinocytes (Figure 4e), confirming the double immunofluorescence results.

To further substantiate the close proximity of epidermal GPR115 and keratins, we performed silver-enhanced immunogold electron microscopy using the GPR115^ECD^ Ab. In normal skin, immunogold particles were present in suprabasal, noncornified keratinocytes, where KRT1/10 represents almost all keratins, in addition to low, residual KRT5/14 (Figure 4f). The label was highly enriched at and restricted to bundles of keratin filaments (Figure 4g). Of note, the distribution of immunogold particles at the filaments showed a remarkable periodicity of ~48 nm (Figure 4h). Given the length of ~45 nm for the coiled-coil backbone of keratins, this is compatible with an association of GPR115 with the KRT1 head or tail domains [24]. Further, groups of silver-enhanced gold particles located along the desmosomes with a distance of ~58 nm to the plasma membrane (Figure 4i), in approximate agreement with the known distance of keratins inserting at the C-terminus of desmoplakin [25]. In sum, GPR115 and KRT1 are highly colocalized inside suprabasal keratinocytes in vivo and in HaCaT keratinocytes, despite GPR115 being a 7TM receptor.

### 3.6. Endogenous GPR115 Is Unglycosylated and Uncleaved in HaCaT Keratinocytes

To gain some insight into principles that might explain this unexpected intracellular localization of GPR115, we turned to biochemical analysis. First, we probed whether endogenous GPR115 extracted from the epidermis and HaCaT keratinocytes represents the naked protein core, as predicted from its keratin-based localization. In total protein extracts, a strong band at ~68 kDa was detected in the epidermis and HaCaT cells, and a weaker one was detected at 75 kDa in HaCaT cells (Figure 5a), which is in reasonable correlation with the calculated molecular weight of GPR115 (NP_722580.3) of 75.3 kDa. The relative molecular weight of GPR115 remained unaltered upon the treatment of HaCaT cells with tunicamycin to block N-linked glycosylation (Figure 5b). Thus, endogenous GPR115 is not N-glycosylated.

To further substantiate this, HA-GPR115 was transfected into Cos-7 cells, which resulted in protein species that migrate as multiple bands at ~90–110 kDa (Figure 5a). Their extractability with RIPA buffer is consistent with that of other aGPCRs [16,17]. The GPR115 expressed in Cos-7 cells is N-glycosylated because, in PNGase F-treated lysates or in lysates of these cells cultured before with tunicamycin, the GPR115 molecular weight was reduced to 65 kDa (Figure 5c).

To examine whether human GPR115 was cleaved at the GPS as with most aGPCRs [10], N-terminal HA- and C-terminal Flag-tagged GPR115 was expressed in Cos-7 cells. Most human GPR115 remained noncleaved (Figure 5d), confirming data obtained from mouse GPR115 [9]. Because the molecular weights of endogenous GPR115 and of the de-N-glycosylated GPR115 expressed in Cos-7 cells are similar, endogenous GPR115 is very likely also not cleaved, arguing that the naked protein core localizes to KRT1/10.

Biochemical analyses of HaCaT WT cells and the GPR115KO clones further confirmed our data (Figure 5e). The amount of GPR115 and KRT1/10 protein decreased simultaneously in all HaCaT GPR115KO clones.

### 3.7. Transfected GPR115 Fails to Colocalize with KRT1 in Cos-7 Cells, Even upon Deglycosylation

One major difference between endogenous and transfected GPR115 resides in the extensive N-glycosylation of the latter. Thus, we asked whether this difference determined either the KRT1 association or the membrane localization. Upon the transfection of Cos-7 cells with a GPR115 variant in which most N-glycosylation sites were mutated (GPR115 mutGlyc, Appendix A), its molecular weight decreased to ~75–80 kDA; when treated with PNGase F, it decreased to 65 kDa (Figure 6a). The mutation of all N-glycosylation sites (Appendix A) decreased the size to 63–65 kDa (Figure 6b). These N-glycosylation mutants lost their membrane localization, and unglycosylated GPR115 showed a diffuse intracellular distribution (Figure 6c,d). This was not surprising, given that Cos-7 cells lack KRT1.

Thus, we transfected HaCaT keratinocytes with HA-tagged WT GPR115, which appeared partly at the cell surface (Figure 6e). The fraction of transfected GPR115 that was localized inside cells did not costain with keratins when using either GPR115 or the HA-tagged antibody (Figure 6f). Further, the localization of transfected GPR115 was similar in HA-GPR115-transfected keratinocytes of WT mice and of mice deficient for type I keratins (*KtyI^-/-^*) (Figure 6g). Thus, transfected GPR115 localized independent of keratins.

To address whether endogenous GPR115 might be associated with intracellular membranes via keratins, we focused on the recent finding that the peripheral endoplasmic reticulum (ER) partly exhibits nanometer proximity to keratin filaments [22]. We stained HaCaT keratinocytes using the GPR115^ECD^ Ab and Abs directed to reticulon 4 (RTN4), a marker for peripheral ER tubules, and to cytoskeleton-associated protein 4 (CKAP4), a marker of ER sheets. These proteins and GPR115 localized in distinctly different patterns (Figure 6h). Thus, the colocalization of endogenous GPR115 and KRT1 most unlikely does not occur along the peripheral ER.

## 4. Discussion

By now, the occurrence of actin in the nucleus, in addition to the cytoplasm, and of β-catenin at adherens junctions and in the nucleus represents textbook knowledge and serves as examples for many other proteins that display noncanonical localization and function [26,27].

Here, we provide strong evidence for the localization of GPR115, an aGPCR, inside keratinocytes in either skin cryosections or cultured HaCaT cells. In agreement with the reanalyzed epidermal scRNAseq data [1], we detected the protein in rare basal and in almost all suprabasal, nonkeratinized keratinocytes of normal skin by immunotechniques applying a carefully validated specific Ab directed to the ECD of this aGPCR. In psoriatic skin, characterized by excessive keratinocyte proliferation and altered differentiation, GPR115 expression was diminished. GPR115 also occurred in a subset of Ki-67-positive suprabasal keratinocytes. These data suggest an involvement of GPR115 in epidermal differentiation.

To further evaluate the roles of GPR115 and keratinocyte differentiation, organotypic cocultures of HaCaT keratinocytes with fibroblasts were established, conditions that induce differentiation and the formation of a stratified epithelium with all features of a normal epidermis [28]. In these constructs, almost all suprabasal keratinocytes coexpressed GPR115 and KRT1/10. Most notably, a CRISPR/Cas9-mediated deletion of *ADGRF4* decreased the number of keratinocyte layers and abrogated *KRT1*. In similar experimental settings, a disruption of *PTTG1* reduced epidermal stratification to the extent that cells appeared as a simple epithelium [1,29]. Disruptions of either *HELLS* or *UHRF1* also resulted in a thinner epidermis and the suppression of epidermal homeostasis [1]. In light of these observations, it is likely that GPR115 not only regulates KRT1 but controls the early stages of epidermal differentiation, which is consistent with its expression in few proliferating basal cells [1]. In support, *Kty1^-/-^* mice showed a near-normal epidermis in which all epidermal layers were maintained, although these mice died a few hours after birth [23].

Our hypothesis that GPR115 is involved in epidermal differentiation is not contradicted by the lack of an overt skin phenotype in *Adgrf4^-/-^* mice [9]. Given that *ADGRF4* clusters tightly with *ADGRF2* and that both are only present in land-living mammals [9], they likely exert redundant functions in mouse skin. To what extent GPR111/*ADGRF2*, nearly absent in human keratinocytes [1], can substitute for GPR115 remains to be analyzed. Interestingly, *Adgrf4^-/-^* mice revealed that this protein plays an important role in enamel mineralization via the regulation of carboanhydrase-6 expression in ameloblasts [30].

The most unexpected finding of our study is the close colocalization of endogenous GPR115 and KRT1/10, which is strongly supported by immunogold electron microscopy. At first glance, such a colocalization appears unusual for a GPCR that is commonly present at and signaling from extra- and intracellular membranes [31,32]. Is it conceivable that only a GPR115 fragment such as the ECD or a part thereof associates alongside KRT1/10 filaments? Although not rigorously disproving this, the fsct that both GPR115 Abs, binding near the N- and C-terminus, respectively, reveal a close keratin association supports the view that intact GPR115 associates with keratins. Moreover, both human and mouse [9] GPR115 are not proteolytically cleaved at the GPS, and they display a similar molecular weight predicted for the nonglycosylated protein species in Western blots. This infers that full-length endogenous GPR115 is associated with KRT1/10.

The noncanonical, intracellular localization of an aGPCR has not been reported until now. At the same time, the noncanonical occurrence of cytoskeletal keratins is widely accepted, as exemplified by nuclear KRT17 and by extracellular KRT19 [33,34]. Given that GPCRs contain hydrophobic 7TM helices, it is unlikely that they exist as freely floating entities inside the cytoplasm but reside in membrane compartments, which is fully consistent with our localization data for transfected but not endogenous GPR115.

Having said this, KRTs 8, 18, and 19 have all been reported to functionally interact with the cystic fibrosis transmembrane regulator (CFTR), a multispan transmembrane receptor. The interaction of KRT8 with the CFTRΔF508 mutant, the most common mutation in patients with cystic fibrosis, at nucleotide-binding domain 1 (NBD1) inhibits the translocation of CFTR to the cell surface [35,36]. A deletion of KRT8 rescued CFTR cell surface localization [37]. KRT18 and KRT19, on the other hand, were shown to stabilize CFTR at the cell surface by direct protein interactions [38,39].

Most recently, epidermal keratins were identified in close association with peripheral ER compartments at desmosomes, where they are crucial to maintaining desmosome-associated ER compartments, as seen by the disruption of ER-desmosome complexes upon the expression of a disease-associated KRT14 mutation [40]. Based on parallel staining of GPR115 and the ER markers RTN4 and CKAP4 in HaCaT keratinocytes, it appears unlikely that the GPR115 species that localize close to KRT1/10 reside in an ER compartment. Further experiments are required to identify the nature of the intracellular, KRT1/10-associated compartment in which GPR115 resides. This should pave the way to understanding the biological significance of GPR115’s association with KRT1/10.

## Figures and Tables

**Figure 1 cells-11-03151-f001:**
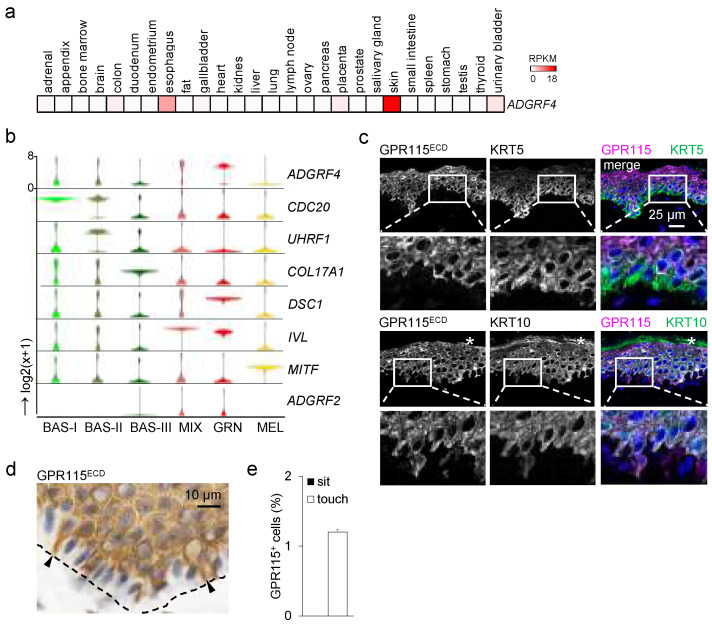
**GPR115/*ADGRF4* is present in rare basal and almost all suprabasal keratinocytes.** (**a**) Transcriptomic profile of *ADGRF4* in human tissues (bulk RNAseq) [11], given as RPKM (reads per kilobase per million mapped reads). (**b**) Reanalysis of scRNAseq data of human epidermis [1]. Violin plots of relative gene expression from basal cell types I-III (BAS), a mixed cluster (MIX, characterizing the transition from type-IV basal cell to spinous and granular keratinocytes), granular keratinocytes (GRN), and melanocytes (MEL). (**c**) Costaining for GPR115 and keratins of normal skin cryosections (star: cornified layer). (**d**) In images of a horse radish peroxidase based immunostained epidermis, few GPR115^+^ basal cells were seen at higher magnification (arrows; broken line: basal membrane). (**e**) Quantitation of GPR115^+^ cells in the basal layer, which either “sit at” or “touch” the basal membrane with extensions (dorsal skin, n = 4 donors; n = 1000–2000 basal cells/donor, mean ± SEM).

**Figure 2 cells-11-03151-f002:**
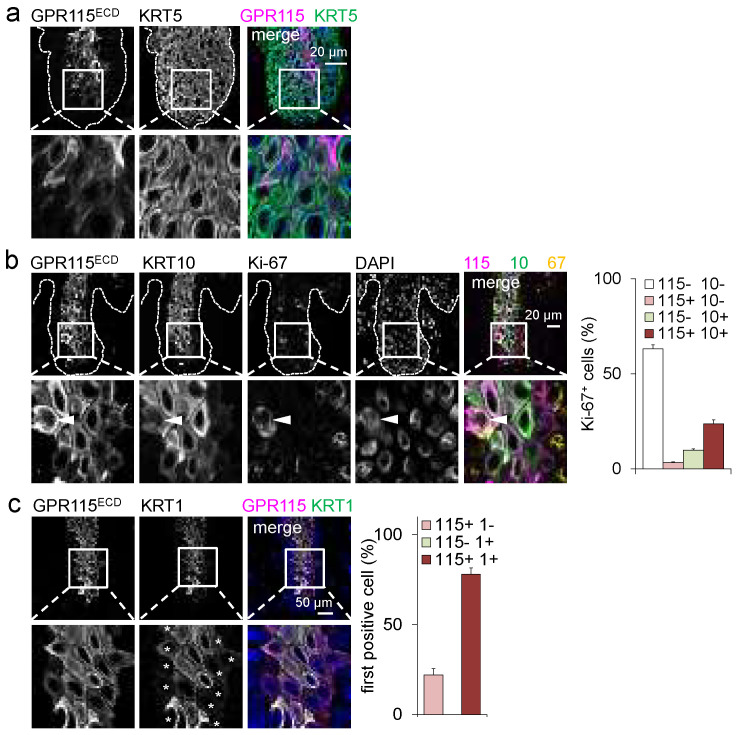
**GPR115 and KRT1/10 were delayed in rete ridges of lesional psoriatic skin.** (**a**) In a rete ridge, suprabasal keratinocytes still expressed KRT5, whereas the emergence of GPR115^+^ keratinocytes was delayed. (**b**) Triple immunostaining of Ki-67, GPR115, and KRT10 revealed a markedly increased proportion of proliferating Ki-67^+^ in the suprabasal layer. Some of these Ki-67^+^ cells expressed GPR115 (arrowhead). The epidermal basal membrane is indicated by a broken line. Right: Quantitation of the percentage of GPR115^+^ cells among Ki-67^+^ suprabasal keratinocytes (n = 10 optical fields, means ± SEM). (**c**) Determination of whether the first stained suprabasal cell is GPR115^+^ and/or KRT1^+^. Left: Costaining; in the insert the rated cells are indicated by asterisks. Right: quantitation of the first GPR115- or KRT1-Ab-stained cells (n = 10 optical fields, means ± SEM).

**Figure 3 cells-11-03151-f003:**
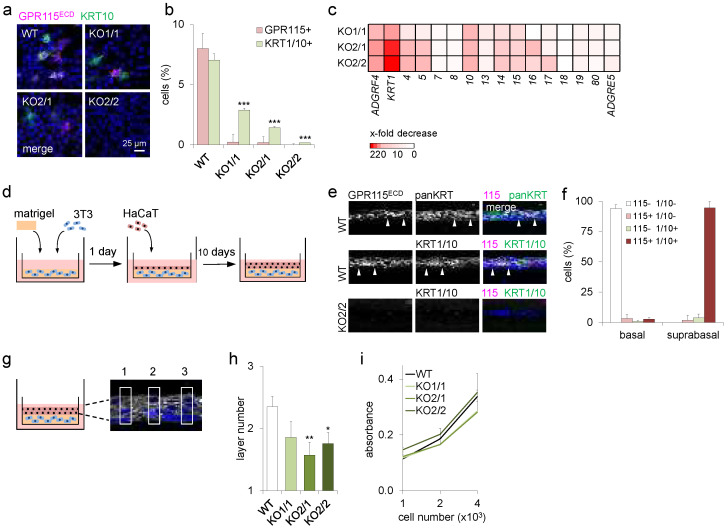
**Loss of *ADGRF4* abrogated *KRT1* and disturbed differentiation in HaCaT keratinocytes.** (**a**) Monolayered HaCaT WT cells and GPR115KO clones were costained for GPR115 and KRT10. (**b**) The percentages of GPR115^+^ and KRT10^+^ cells were determined in each clone (n = 10 optical fields, n = 70–100 cells/field, means ± SEM, *** *p* < 0.001 compared with WT). (**c**) RNA sequencing of HaCaT WT and GPR115KO clones. The ratio of FKPM values in clones and WT is given as an x-fold decrease. (**d**) Scheme generating organotypic skin constructs consisting of Matrigel-embedded 3T3 fibroblasts overlayered with HaCaT WT or GPR115KO cells. (**e**) Cross-cryosections of constructs built with HaCaT WT cells (upper panels) and the GPR115KO2/2 clone (lower panel) were costained for GPR115 and keratins. (**f**) The percentages of GPR115^+^ and KRT1/10^+^ cells were determined in these constructs built with HaCaT WT cells (n = 10 optical fields, n = 20–27 cells/field, means ± SEM). (**g**) Scheme of the determination of the HaCaT layer number. The number of DAPI^+^ nuclei was counted in 10 uniform segments; the distances between the segments were equal (three segments are shown). (**h**) Quantitation of the HaCaT layer number in organotypic skin constructs (n = 5–8 experiments/WT or GPR115KO clone, 3 cross sections/experiment, 10 segments/cross section, means ± SEM, * *p* < 0.05, ** *p* < 0.01 compared with WT). (**i**) Crystal violet assay of 1 × 10^3^, 2 × 10^3^, and 4 × 10^3^ seeded HaCaT WT and GPR115KO cells. Attached cells over time were quantified. Absorbance was measured after 48 h at 590 nm (n = 3 experiments, n = 3 replicates/experiment; means ± SEM).

**Figure 4 cells-11-03151-f004:**
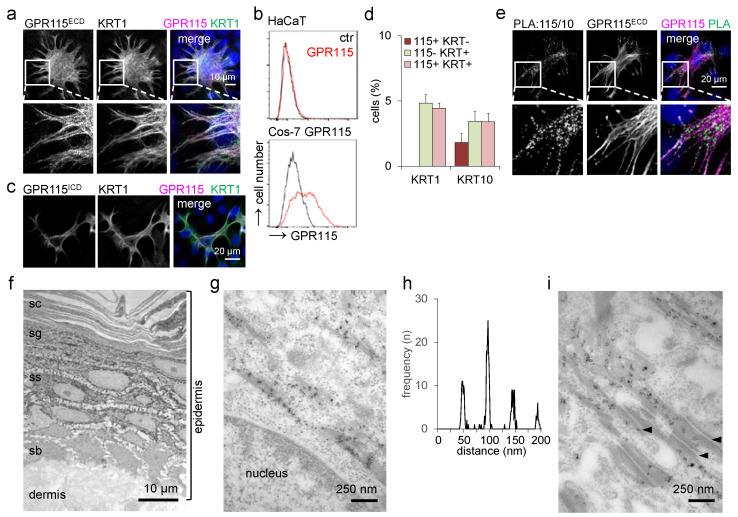
**Endogenous GPR115 colocalizes with KRT1/10.** (**a**) GPR115 and KRT1 costaining of HaCaT cells. The insert illustrates the obvious colocalization and the filament-like, partly dotted GPR115 staining. (**b**) HaCaT WT cells and Cos-7 cells, transfected with GPR115 pcDNA3.1, were cell-surface-stained with the GPR115^ECD^ Ab and analyzed by flow cytometry. (**c**) GPR115^ICD^ and KRT1 Ab costaining of HaCaT cells. (**d**) Calculation of the percentage of GPR115^ECD^-labeled HaCaT cells costained with the KRT1 or KRT10 Abs (n = 10 optical fields, n = 220–320 cells/field, means ± SEM). (**e**) Proximity ligation assay (PLA) applying the rabbit GPR115^ECD^ and mouse KRT10 Abs in HaCaT WT cells. After visualization of the PLA interaction dots, the cells were stained with a fluorophore-labeled antirabbit secondary Ab to visualize GPR115. (**f**–i) GPR115^ECD^ Ab immunogold labeling of normal skin via electron microscopy. (**f**) Immunogold particles were present in all suprabasal, nonkeratinized epidermal layers (sb: stratum basale, ss: stratum spinosum, sg: stratum granulosum, sc: stratum corneum). (**g**) The immunogold particles were located at keratin filaments and grouped with defined distances from each other. (**h**) These distances were quantified in pictures; the frequencies of a certain distance between the particle groups are given. The distances between groups were multiples of 48 nm (48.8 ± 0.4, 96.2 ± 0.3, 144.8 ± 0.4, 193.1 ± 0.5 nm; n = 16 figures, 10–45 particle groups at filaments/figure, means ± SEM). (**i**) Alongside desmosomes (arrowheads), immunogold particle groups were also located at filaments. The distance of such groups to the desmosomes was 57.6 ± 1.3 nm (n = 8 pictures, 2–6 particle groups/picture, means ± SEM).

**Figure 5 cells-11-03151-f005:**
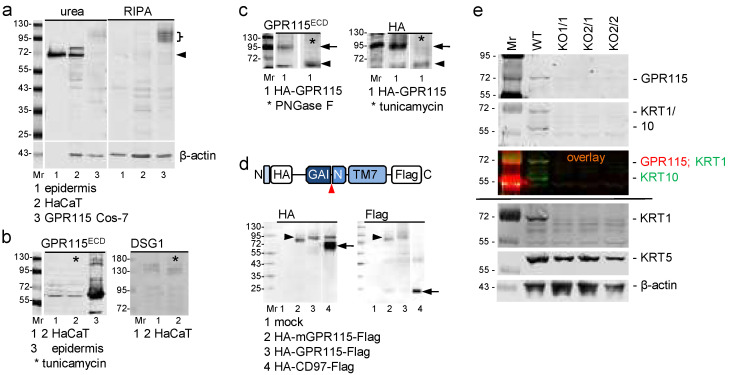
**Endogenous GPR115 is not N-glycosylated and is likely not cleaved at the GPS.** (**a**) Western blot analysis of epidermis, HaCaT, and HA-GPR115 Cos-7 cells in urea-based (**left**) and detergent-based (**right**) lysates applying the GPR115^ECD^ Ab. Only 1 μg of epidermal protein/lane was blotted, while 40 μg/lane of the other lysates were blotted. In the lower part, the loading control using a β-actin Ab is shown. It is negative for the epidermis because only 1 μg of protein was applied. (**b**) Western blot analysis of lysates (urea buffer) of epidermis and HaCaT cells; two different blots are shown. Tunicamycin (*) did not decrease the molecular weight of endogenous GPR115 (**left**) but decreased that of desmoglein 1 (DSG1, **right**), which served as a positive control. (**c**) Western blot analysis of lysates of HA-GPR115 Cos-7 cells treated with PNGase F (**left**) or of HA-GPR115 Cos-7 cells cultured with tunicamycin (**right**); the Abs used are indicated. Most transfected GPR115 is N-glycosylated (arrows); deglycosylation reduced the molecular weight to 65 kDA (arrowheads). (**d**) Lysates of Cos-7 transfected with constructs encoding N-terminal HA- and C-terminal Flag-tagged mouse or human GPR115 and human CD97 were analyzed with the indicated Abs by Western blot; the coding part of the constructs is shown schematically above (arrowhead: putative GPS). In the blots, arrowheads indicate uncleaved full-length human GPR115, and the arrows indicate the N- and C-terminal fragments of cleaved CD97 (positive control). Mouse GPR115, a little smaller than human GPR115 also was not cleaved. (**e**) Western blot analysis of urea-based lysates of HaCaT WT cells and the various GPR115 clones. The applied Abs are indicated on the right. Upper part: First, the blot was consecutively incubated with the rabbit GPR115^ECD^ and mouse KRT1/10 primary Abs. Afterwards both fluorophore-labeled secondary Abs were applied; pictures taken from this blot are shown (the single Abs in grey and their overlay colored). Lower part: same blot after stripping and restaining; the stripped blot was horizontally cut, and the three parts were incubated with either the KRT1, KRT5, or β-actin Abs.

**Figure 6 cells-11-03151-f006:**
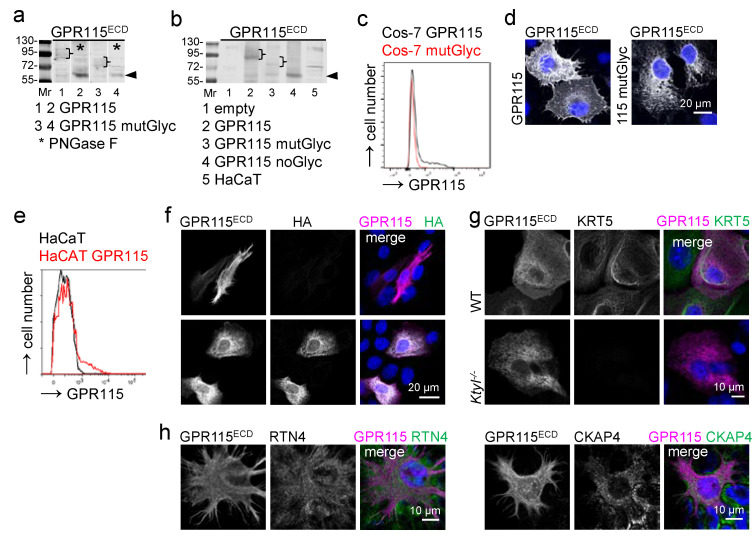
**GPR115 transfected into Cos-7 cells is not keratin-associated.** (**a**) Western blot analysis of Cos-7 cells transfected with the various indicated GPR115 pcDNA3.1 constructs. Mutation of seven potential GPR115 N-glycosylation sites (GPR115 mutGlyc) reduced the molecular weight of transfected GPR115 from ~90–110 to ~75–85 kDa (brackets). PNGase F treatment (*) of the cellular lysates resulted in a ~65 kDa band (arrowhead); thus, further potential GPR115 N-glycosylation sites were used. (**b**) Consistently, mutation of all potential GPR115 N-glycosylation sites (GPR115 noGlyc) reduced the molecular weight to ~63–65 kDa (arrowhead). (**c**,**d**) Cos-7 cells, transfected with GPR115 or GPR115 mutGlyc pcDNA3.1, were stained for GPR115. Mutant GPR115 mainly disappeared from the cell surface, as seen in flow cytometry (**c**) and in stained monolayered cells (**d**). (**e**–**g**) HaCaT WT cells were transfected with HA-GPR115 pcDNA3.1. (**e**) These cells were stained with the GPR115^ECD^ Ab and compared with nontransfected cells in flow cytometry. (**f**) The transfected HaCaT cells were costained using the GPR115^ECD^ and HA Abs to differentiate between endogenous (HA-) and transfected (HA+) GPR115. (**g**) Mouse WT and keratin type I-deficient (*KtyI^-/-^*) keratinocytes were transfected with HA-GPR115 pcDNA3.1 and costained for KRT5 and GPR115; the GPR115^ECD^ Ab labeled only human GPR115. The staining pattern of transfected GPR115 was similar in WT and *KtyI^-/-^* keratinocytes. (**h**) HaCaT WT cells were costained for GPR115 with the ER tubule marker RTN4 and the ER sheet marker CKAP4.

## Data Availability

The data presented in this study are included in this published article or are available from the corresponding author upon reasonable request.

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
