# Peer review of "The Adhesion G-Protein-Coupled Receptor GPR115/ADGRF4 Regulates Epidermal Differentiation and Associates with Cytoskeletal KRT1"

_cells, 2022, doi:10.3390/cells11193151_

Round 1
Reviewer 1 Report
Winkler et al., explore the roles of GPR115 in both regulating keratin 1 (and perhaps regulating differentiation in general) and in associating with KRT1 in vivo and in vitro. They first find that the gene encoding GPR115 is expressed to highest levels predominantly in the suprabasal layers of the skin. They then determine its expression pattern within psoriatic skin rete ridges where it appears earlier than KRT1. Next using WT and GSPR115 knockout HaCat cells in a skin organotypic culture they show that GSPR115 knockout cells do not express the suprabasal keratins 1 and 10 which probably results from loss of skin differentiation. Surprisingly, they go on to show that GPR115 associates with KRT1 fibers in the skin. Further, they explore the role of N-glycosylation in the localization.
This is a novel finding which will be of great interest to the readers of Cells.
I would like the authors to address some questions:
1. In psoriatic skin, you focused on rete ridges. Did you see similar effects as you report for human tissues in Fig. 1C in regions outside of the rete ridges?
2. You find that loss of ADGRF4 decreases the number of induced cell layers in organotypic cultures but that the cell numbers did not change. How do you account for this observation? Where did those cells go? Were changes in cell size involved? Did this result in changes in cell morphology or the number of cell:cell junctions?
3. Were genes such as p63 which are involved in epithelial stratification affected by loss of GPR115?
4. Your finding that GPR115 can regulate the expression of KRT1 but also can bind to the protein is an interesting concept. Could you speculate on how you envision that this occurs? Is this possibly part of a feedback loop where binding to the keratin might prevent GPR115 from interacting with a ligand that would induce upregulation of differentiation including keratin expression?
Author Response
Response to Reviewer 1 comments
- In psoriatic skin, you focused on rete ridges. Did you see similar effects as you report for human tissues in Fig. 1C in regions outside of the rete ridges?
In psoriatic skin, prolonged KRT5 and postponed GPR115 and KRT1/10 expression is best seen in rete ridges. They contain mainly the first suprabasal spinous keratinocytes. In granular keratinocytes, that is outside rete ridges in the granular layer, GPR115 and KRT1 are always present but appear non-homogenously inside the cells with high agreement.
We added this point to the results.
- You find that loss of ADGRF4 decreases the number of induced cell layers in organotypic cultures but that the cell numbers did not change. How do you account for this observation? Where did those cells go? Were changes in cell size involved? Did this result in changes in cell morphology or the number of cell:cell junctions?
The proliferation assay was performed using 2D monolayer cultures, whereas the epidermal layer number was quantified in 3D organotypic cultures. Of course, not only the epidermal layer number, but also the number of epidermal cells is higher when applying HaCaT WT compared to GPR115KO cells. That means HaCaT WT cells proliferate more compared to the GPR115KO clones under organotypic but not under monolayer conditions.
We clarified this point precisely and rewrote this part in “Results”:
- Were genes such as p63 which are involved in epithelial stratification affected by loss of GPR115?
The transcription factor p63 is considered as one master regulator of epidermal differentiation [1].
The expression levels of TP63 mRNA in HaCaT WT cells and of the various HaCaT GPR115Ko clones are similar under monolayer conditions. Transcriptomic profiling of these cells under organotypic conditions will clarify whether genes known to be involved in epidermal stratification and differentiation are regulated after loss of ADGRF4.
- Your finding that GPR115 can regulate the expression of KRT1 but also can bind to the protein is an interesting concept. Could you speculate on how you envision that this occurs? Is this possibly part of a feedback loop where binding to the keratin might prevent GPR115 from interacting with a ligand that would induce upregulation of differentiation including keratin expression?
Thanks for the interesting idea that binding to KRT1 modifies the function of GPR115. Although we have no evidence so far for a direct interaction between the two proteins, our combined data clearly indicate that the presence of KRT1 results from GPR115 activity and not vice versa. We point out that GPR115 is an orphan receptor for which no interaction or binding partners are presently known. Having said this, it does not exclude that both intracellular and the transfected membrane-bound variant could signal. This vital issue requires further experiments.
Transfected GPR115, appearing at the cell surface, is capable of signaling. In aGPCRs, the juxtamembraneous part right after the potential GPS, not functional in GPR115, carries a tethered agonistic sequence [2,3]. Tethered agonist-derived peptides can activate the respective receptor. Peptides derived from the putative GPR115 tethered agonist sequence were not active in second messenger assays with transfected GPR115. However, in HEK293 cells, transfected human GPR115 raised inositol 1-phosphate (IP1) upon stimulation with a GPR116-derived peptide [4], and transfected mouse GPR115 was constitutively active when co-transfected with Gα15 [5].
In view of the involvement of keratins K8 and K18 in trafficking of CFTR ΔF508, a similar role of quality control function for KRT1/10 could be envisaged, along either the biosynthetic or the degradative route. Future studies in KRT1 KO cells should be able to resolve this. Furthermore, association of GPR115 and KRT1 may contribute to limit the activity of the former during terminal differentiation in the epidermis. To address this, we are currently investigating whether the two proteins directly interact with each other. Finally, in view of the major role of KRT1 in cornified envelope formation [6], GPR115 might assist in preparing KRT1/10 filaments for their association with components of the cornified/lipid envelope.
Reviewer 2 Report
The manuscript by Winkler et al. describes an association between GPR115 and KRT1. The manuscript is clearly written and the experimental plan is well described and designed. Although the reported experiments show an intracellular localization of GPR115 in close proximity of KRT1/10 it is still not clear where, or in which compartment, the 7-TM protein is actually present. The authors report that staining with RTN-4 suggests that GPR115 is not associated with the ER. We suggest to test other ER markers to confirm this hypothesis. In addition, a co-immunoprecipitation or in vitro binding assay may help in understanding whether GPR115 interacts with KRT1/10 or whether the two proteins are only close one to each other.
Minor point:
Lines 188-189: the sentence "using HaCaT wild-type (WT) keratinocytes (as a negative con- trol their endogenous ADGRF4 was deleted)" is not clear. We would suggest to replace it with ""using HaCaT wild-type (WT) keratinocytes and a HaCaT knockout clone called KO 2/2, where the endogenous ADGRF4 was deleted"
Author Response
Response to Reviewer 2 Comments
- The authors report that staining with RTN-4 suggests that GPR115 is not associated with the ER. We suggest to test other ER markers to confirm this hypothesis.
The ER forms an intricate meshwork that is composed of interconnected tubules and sheets. We labelled the ER sheets with an antibody directed to cytoskeleton-associated protein 4 (CKAP4, CLIMP-63). In HaCaT keratinocytes, the CKAP4 staining pattern completely differs from that of GPR115 and KRT1.
We added CKAP4 staining to figure 6h.
- In addition, a co-immunoprecipitation or in vitro binding assay may help in understanding whether GPR115 interacts with KRT1/10 or whether the two proteins are only close one to each other.
To clarify whether GPR115 and KRT1 directly interact is outside the focus of our present work. It will take time and needs models to test it. Most important, we do not have an in vitro model that mimics the endogenous situation. Transfected GPR115 does not associate with endogenous KRT1 in keratinocytes. Further, transfected KRT1 and KRT10 do not form filaments as do transfected KRT5 and KRT14. KRT1 is nearly insoluble. It is soluble only in solutions containing high molar urea, conditions that prevent protein interactions. Working with parts of KRT1 using tagged head-, rod- and tail-encoding constructs will help us. However, doing these experiments will take time and is a new story.
- Minor point:
Lines 188-189: the sentence "using HaCaT wild-type (WT) keratinocytes (as a negative con- trol their endogenous ADGRF4 was deleted)" is not clear. We would suggest to replace it with ""using HaCaT wild-type (WT) keratinocytes and a HaCaT knockout clone called KO 2/2, where the endogenous ADGRF4 was deleted"
We changed this.
Reviewer 3 Report
This is an excellent paper describing how the adhesion G protein-coupled receptor GPR115/ADGRF4 2 regulates epidermal differentiation. I only have a few minor points to be addressed before MS publication.
Please add a high-zoom image for the GPR115 expression with KART in Figure 1C and Figure 2A.
Please replace the blot for beta actin for a better image in Figure 5E.
3-Overall, this study presented very nice, organized, and clear figures supporting their results.
Author Response
Response to reviewer 3 comments
1. Please add a high-zoom image for the GPR115 expression with KART in Figure 1C and Figure 2A.
We added the high-zoom images.
2. Please replace the blot for beta actin for a better image in Figure 5E.
This is the same blot after GPR115 and KRT1/10 detection, stripping and re-staining. The stripped blot was horizontally cut and the lowest part was incubated with the β-actin Ab. Thus, we could not replace the blot.
Round 2
Reviewer 2 Report
The manuscript "The adhesion G protein-coupled receptor GPR115/ADGRF4 regulates epidermal differentiation and associates with cytoskeletal KRT1" by Winkler et al. has been revised according to my previous suggestions. I appreciate staining with a novel ER resident protein; however a co-staininig should be more informative on GR115 localization also because the cell in panel 6-h (stained for GR115) looks quite different form those reported in panels stained with anti RTN4 or CKAP4. Please provide, if possible, a co-staining image.
I understand the difficulties in performing co-immunoprecipitation or in vitro interaction studies and accept your response.
Author Response
Response to Reviewer 2 comments
The manuscript "The adhesion G protein-coupled receptor GPR115/ADGRF4 regulates epidermal differentiation and associates with cytoskeletal KRT1" by Winkler et al. has been revised according to my previous suggestions. I appreciate staining with a novel ER resident protein; however a co-staininig should be more informative on GR115 localization also because the cell in panel 6-h (stained for GR115) looks quite different form those reported in panels stained with anti RTN4 or CKAP4.
Please provide, if possible, a co-staining image.
We co-stained HaCaT WT cells for GPR115/RTN4 and for GPR115/CKAP4. The mouse monoclonal RTN4 Ab is not really good.
I understand the difficulties in performing co-immunoprecipitation or in vitro interaction studies and accept your response.